Proceedings of the 7th Symposium on Advances in Approximate Bayesian Inference, 2025 1–26

# Sparse Gaussian Neural Processes

**Tommy Rochussen**[1,2]                                                    TOMMY.ROCHUSSEN@TUM.DE

**Vincent Fortuin**[1,2,3]

[1]*Helmholtz AI*

[2]*Technical University of Munich*

[3]*Munich Center for Machine Learning*

## Abstract

Despite significant recent advances in probabilistic meta-learning, it is common for practitioners to avoid using deep learning models due to a comparative lack of interpretability. Instead, many practitioners simply use non-meta-models such as Gaussian processes with interpretable priors, and conduct the tedious procedure of training their model from scratch for each task they encounter. While this is justifiable for tasks with a limited number of data points, the cubic computational cost of exact Gaussian process inference renders this prohibitive when each task has *many* observations. To remedy this, we introduce a family of models that meta-learn sparse Gaussian process inference. Not only does this enable rapid prediction on new tasks with sparse Gaussian processes, but since our models have clear interpretations as members of the neural process family, it also allows manual elicitation of priors in a neural process for the first time. In meta-learning regimes for which the number of observed tasks is small or for which expert domain knowledge is available, this offers a crucial advantage.

## 1. Introduction

In many high-stakes environments where uncertainty estimates are key to making real-world decisions, obtaining well-calibrated predictive distributions is of vital importance. While many models have been developed that can produce such probabilistic predictions, it is often the case that predictions are required for multiple related tasks, such that it would be desirable to have a probabilistic model that can make rapid predictions on new tasks without the need for task-specific training. Such is the case in the probabilistic meta-learning paradigm. While meta-learning has received an abundance of attention from the research community over the last decade (Finn et al., 2017; Gordon et al., 2019; Hospedales et al., 2022), the most notable class of *probabilistic* meta-model is, without doubt, the neural process family (NP; Garnelo et al., 2018a,b; Dubois et al., 2020). Recent advances in NPs have led them to reach astonishing heights in performance, representing the state-of-the-art in data-based approaches to weather and climate modeling (Bodnar et al., 2024; Allen et al., 2025; Ashman et al., 2024b), for example. Despite such impressive performance, industry practitioners seldom opt for deep learning models owing to their inherent lack of interpretability (Li et al., 2022), and instead prefer more traditional approaches such as kernel methods (Hofmann et al., 2008) that are easier to explain to non-technical stakeholders, even if they are incapable of meta-learning.

Perhaps the most ubiquitous probabilistic model that practitioners turn to is the Gaussian process (GP; Rasmussen and Williams, 2005). With GPs, users can leverage their domain expertise to specify meaningful priors with which to bias predictions, any free parameters tend to have clear interpretations, and schemes such as automatic relevance

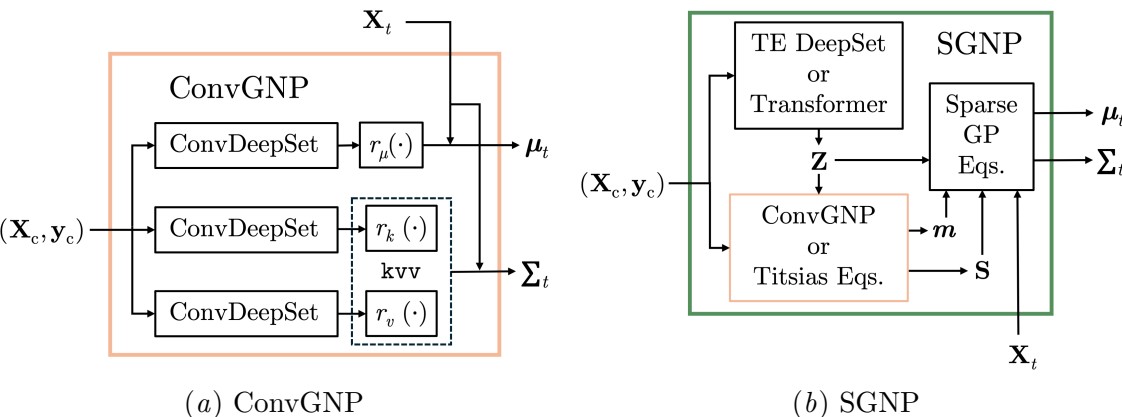

(a) ConvGNP                    (b) SGNP

Figure 1: Computational diagrams of the ConvGNP (Markou et al., 2022) and the SGNP (ours). $\boldsymbol{\mu}_t$ and $\boldsymbol{\Sigma}_t$ denote the mean and covariance matrix that parameterise the multivariate Gaussian predictive distribution over the target outputs, $\mathbf{y}_t$.

determination (ARD) return useful information about the data. Unfortunately, evaluating the predictive distribution has a computational complexity of $\mathcal{O}(n^3)$ in the number of data points, meaning practitioners must resort to scalable Gaussian process approximations (Quiñonero-Candela and Rasmussen, 2005; Snelson and Ghahramani, 2005) when their dataset is large. Sparse GPs are typically trained via variational inference (Jordan et al., 1999; Blei et al., 2016), meaning that even if there are ways to meta-learn hyperparameter selection across tasks, their variational parameters need to be optimised from scratch for each new task. This inflexibility can be a major problem, especially when practitioners require accurate probabilistic predictions as soon as possible after observing a new task.

Motivated to find a way to meta-learn sparse variational GP inference, we develop an approach that leverages amortised variational inference in a *per-dataset* way, similar to how amortised inference is performed in NPs (Garnelo et al., 2018b). To achieve this, we use permutation-invariant set functions to map from datasets to the variational parameters of a sparse GP. These parameters define the posterior distribution over a sparse representation of the dataset, and since this representation is then processed along with the test inputs (albeit via exact Bayesian inference) to make predictions, the model can be classed as a neural process. We therefore refer to our class of model as the *Sparse Gaussian Neural Process*. We summarise our contributions below.

1. We introduce a new type of neural process that is interpretable, data efficient, and scalable.

2. We develop variants of the model that are particularly well-suited to either regression or classification.

3. We show, on synthetic and real-world data, that our model outperforms existing neural processes when the number of observed tasks is small or when accurate prior knowledge is available.

## 2. Background

### 2.1. Sparse Variational Gaussian Processes

Here, we review the variational approach to sparse GP inference as developed in Titsias (2009) and Hensman et al. (2013, 2015). We denote the collection of $n$ input data points of dimensionality $d$ by $\mathbf{X} \in \mathbb{R}^{n \times d}$ and the corresponding outputs by $\mathbf{y} \in \mathbb{R}^n$. Our modelling assumptions are that the outputs were generated by sampling a latent function $f(\cdot)$ from some Gaussian process, evaluating the sampled function at the inputs to obtain the vector $\mathbf{f}$, and passing the function values through some noise model that is appropriate to the task. For regression, this noise model might be independent additive Gaussian noise of standard deviation $\sigma_y$:

$$\mathbf{y} = \mathbf{f} + \sigma_y \boldsymbol{\epsilon}, \quad \epsilon_i \sim \mathcal{N}(0, 1) \tag{1}$$

whereas for classification the noise model might be independent binary noise that follows a Bernoulli distribution:

$$\mathbf{y} = \boldsymbol{\epsilon}, \quad \epsilon_i \sim \mathcal{B}\big(\sigma(f_i)\big) \tag{2}$$

where $\sigma(\cdot)$ denotes a sigmoidal inverse-link function. Regardless of the noise model, it is used to define the likelihood $p(\mathbf{y}|\mathbf{f}) = \prod_i p(y_i|f_i)$. Having specified an appropriate covariance (kernel) function $k(\cdot, \cdot)$ for the problem, we can construct the data covariance matrix $\mathbf{K_{ff}}$ and define the prior over the values of the latent function $p(\mathbf{f}) = \mathcal{N}(\mathbf{f}; \mathbf{0}, \mathbf{K_{ff}})$. The likelihood and prior can be used to define a posterior distribution over the latent function values using Bayes' theorem:

$$p(\mathbf{f}|\mathbf{y}) = \frac{p(\mathbf{f})p(\mathbf{y}|\mathbf{f})}{p(\mathbf{y})}. \tag{3}$$

However, the posterior is only analytically tractable if the likelihood is Gaussian, and even then, it requires $\mathcal{O}(n^3)$ operations, leaving it computationally intractable if the dataset is too large.

Instead, we introduce a set of $m \ll n$ *inducing points* $\mathbf{Z} \in \mathbb{R}^{m \times d}$ that live in the same space as $\mathbf{X}$, and whose job it is to summarise the input data. The latent function values at these points are denoted as $\mathbf{u} \in \mathbb{R}^m$ and are referred to as inducing *outputs*. We define a variational posterior over the inducing outputs $q(\mathbf{u}) := \mathcal{N}(\mathbf{u}; \mathbf{m}, \mathbf{S})$ where $\mathbf{m} \in \mathbb{R}^m$ and $\mathbf{S} \in \mathbf{S}_+^m$ are the variational parameters that will be, alongside $\mathbf{Z}$, found via optimisation[1]. Our goal is to find the inducing points $\mathbf{Z}$ and approximate posterior $q(\mathbf{u})$ that allow us to best approximate the predictions we would obtain if we could perform exact inference. More precisely, we select the parameters which minimise the Kullback-Leibler divergence $\mathrm{KL}[q(\mathbf{f}, \mathbf{u})\|p(\mathbf{f}, \mathbf{u}|\mathbf{y})]$ (Titsias, 2009) which is achieved by maximising the evidence lower bound (ELBO) (Hensman et al., 2015):

$$\log p(\mathbf{y}) \geq \sum_{i=1}^{n} \mathbb{E}_{q(f_i)}[\log p(y_i|f_i)] - \mathrm{KL}[q(\mathbf{u})\|p(\mathbf{u})] \tag{4}$$

$$:= \mathcal{L}_{\mathrm{ELBO}}(\mathcal{D}). \tag{5}$$

Since this objective is a lower bound to the log marginal likelihood, it can be used to perform model selection simultaneously. Furthermore, the bound can be estimated from a batch of

---

1. Note that we use $\mathbf{S}_+^m$ to denote the set of symmetric $m \times m$ positive semi-definite matrices.

$b \ll n$ data points, making it amenable to stochastic variational inference (Hofmann et al., 2008). Note that, if the likelihood is Gaussian and we are not interested in minibatching, the optimal approximate posterior takes the form

$$q^*(\mathbf{u}) = \mathcal{N}(\mathbf{u}; \mathbf{m}^*, \mathbf{S}^*) \tag{6}$$

where $\mathbf{m}^* = \sigma_y^{-2}\mathbf{K_{uu}}\mathbf{\Sigma}\mathbf{K_{uf}}\mathbf{y}$, $\mathbf{S}^* = \mathbf{K_{uu}}\mathbf{\Sigma}\mathbf{K_{uu}}$, $\mathbf{\Sigma} := (\mathbf{K_{uu}} + \sigma^{-2}\mathbf{K_{uf}}\mathbf{K_{fu}})^{-1}$, and $\mathbf{K_{uu}}$ is the inducing point covariance matrix (Titsias, 2009).

Having trained the sparse GP, we predict the function values at the test points $\mathbf{f}_t$ by approximating the posterior distribution

$$p(\mathbf{f}_t|\mathbf{y}) = \int p(\mathbf{f}_t|\mathbf{u})p(\mathbf{u}|\mathbf{y})\mathrm{d}\mathbf{u} \tag{7}$$

$$\approx \int p(\mathbf{f}_t|\mathbf{u})q(\mathbf{u})\mathrm{d}\mathbf{u} \tag{8}$$

$$= \mathcal{N}\left(\mathbf{f}_t; \mathbf{Am}, \mathbf{K_{f_t f_t}} + \mathbf{A}(\mathbf{S} - \mathbf{K_{uu}})\mathbf{A}^\top\right) \tag{9}$$

where $\mathbf{K_{f_t f_t}}$ denotes the test data covariance matrix, and $\mathbf{A} := \mathbf{K_{f_t u}}\mathbf{K_{uu}^{-1}}$ (Hensman et al., 2015).

## 2.2. Neural Processes

Given $n_c$ context observations from a task $\mathcal{D}_c = \{\mathbf{X}_c, \mathbf{y}_c\}$ where $\mathbf{X}_c \in \mathbb{R}^{n_c \times d}$ and $\mathbf{y}_c \in \mathbb{R}^{n_c}$, suppose we would like to obtain the posterior predictive distribution $p(\mathbf{y}_t|\mathcal{D}_c, \mathbf{X}_t)$ over the outputs $\mathbf{y}_t \in \mathbb{R}^{n_t}$ corresponding to a collection of $n_t$ target inputs $\mathbf{X}_t \in \mathbb{R}^{n_t \times d}$. After training on a meta-dataset $\Xi = \{\mathcal{D}_c^{(j)}, \mathcal{D}_t^{(j)}\}_{j=1}^{|\Xi|}$ of observations from distinct tasks, NPs are capable of meta-learning such predictive inference for new tasks. All NPs consist of an encoder $e(\cdot)$ and a decoder $d(\cdot)$, but there is variation in how the encoding is treated. The encoder is a neural set function that maps from a context set $\mathcal{D}_c$ to an approximate posterior over a task-specific latent variable $\mathbf{z} \in \mathbb{R}^r$:

$$q(\mathbf{z}|\mathcal{D}_c) = p\big(\mathbf{z}|e(\mathcal{D}_c)\big). \tag{10}$$

This latent variable serves as a fixed-length representation of the context dataset, and its purpose is to capture the essence of the task via the context set. If it is assumed to be deterministic—i.e., that the approximate posterior is collapsed to a delta function—then the NP is called a *conditional* NP (Garnelo et al., 2018a) and the latent variable is denoted by $\mathbf{r}$. Otherwise, the NP is called a *latent* NP. The decoder is a mapping from the latent variable and the target inputs to a conditional distribution over the target outputs:

$$p(\mathbf{y}_t|\mathbf{X}_t, \mathbf{z}) = p\big(\mathbf{y}_t|d(\mathbf{X}_t, \mathbf{z})\big). \tag{11}$$

While this distribution is homoscedastic in a latent NP, it is heteroscedastic in a conditional NP. The predictive distribution is then obtained by marginalising out the latent variable:

$$p(\mathbf{y}_t|\mathcal{D}_c, \mathbf{X}_t) = \int q(\mathbf{z}|\mathcal{D}_c)\, p(\mathbf{y}_t|\mathbf{X}_t, \mathbf{z})\, \mathrm{d}\mathbf{z} \tag{12}$$

which can be approximated via Monte Carlo integration. Due to the sifting property of the delta function, the predictive distribution simplifies to

$$p(\mathbf{y}_t|\mathcal{D}_c, \mathbf{X}_t) = p(\mathbf{y}_t|\mathbf{X}_t, \mathbf{r}), \quad \mathbf{r} = e(\mathcal{D}_c) \tag{13}$$

in the case of a conditional NP. Due to the intractable predictive distribution in a latent NP, training is performed by optimising the ELBO defined as:

$$\log p(\mathbf{y}_t|\mathcal{D}_c, \mathbf{X_t}) \geq \mathbb{E}_{q(\mathbf{z}|\mathcal{D})}[\log p(\mathbf{y}_t|\mathbf{X}_t, \mathbf{z})] - \mathrm{KL}[q(\mathbf{z}|\mathcal{D})\|q(\mathbf{z}|\mathcal{D}_c)] \tag{14}$$

averaged over all tasks, where $\mathcal{D}$ denotes the union of context and target observations (Dubois et al., 2020). In a conditional NP, training is performed simply by maximising the predictive likelihood of the target observations. Note that the context set is typically a random strict subset of the data $\mathcal{D}_c \subset \mathcal{D}$, while the target set is the full collection $\mathcal{D}_t = \mathcal{D}$.

## 2.3. Neural Set Functions

For a function to be a valid set function, it must be permutation invariant with respect to its inputs (Zaheer et al., 2017). A further consideration is for it to be able to handle sets of differing sizes.

**DeepSets.** The DeepSet (Zaheer et al., 2017) works by passing each of the $n$ observations $(\mathbf{x}_c^{(i)}, y_c^{(i)})$ in a dataset through a multilayer perceptron (MLP) to obtain pointwise embeddings $\mathbf{r}^{(i)}$, and then taking the mean of the embeddings to produce a representation of the set of observations $\mathbf{r} = \frac{1}{n}\sum_i \mathbf{r}^{(i)}$. The set representation can then be used for arbitrary downstream tasks. In a classic conditional NP (Garnelo et al., 2018b), a DeepSet is used as the encoder[2], and the set representation is then concatenated with a target location and passed through another MLP to obtain a target prediction conditioned on the context set (Garnelo et al., 2018a). Note that it is the use of averaging that makes a DeepSet both permutation invariant (summing would also satisfy this) and invariant to the set size (summing would not satisfy this).

**ConvDeepSets.** With the goal of developing a translation-equivariant NP, Gordon et al. (2020) introduce the ConvDeepSet. Instead of obtaining a fixed-length vector $\mathbf{r}$ to represent a dataset, a ConvDeepSet returns a function $r(\cdot)$. In the context of NPs, a ConvDeepSet's functional representation can then be queried directly at the target locations to obtain predictions. Importantly, a translation of the dataset results in an identical translation of the functional representation. A ConvDeepSet operates on datasets via

$$\Phi(\mathcal{D}) = \rho\big(E(\mathcal{D})\big), \quad E(\mathcal{D}) = \sum_i \phi(y_i)\psi(\cdot - \mathbf{x}_i) \tag{15}$$

where $\psi$ is the squared-exponential kernel with a learnable lengthscale, $\phi$ is a power series of order one (assuming there is just one output per input in the dataset; see Gordon et al.,

---

2. In a classic latent NP, a DeepSet is also used as the encoder, but the set representation is used to parameterise a distribution over the latent variable rather than being used as the (deterministic) latent variable directly.

2020) such that $\phi(y_i) = [1, y_i]^\top$, and $\rho$ is a convolutional neural network (CNN). The two channels that $\phi$ outputs are termed the *density* and *data* channels respectively. The intuition behind the need for the density channel is to indicate to the model where data have been observed, for example to distinguish between an absent point and an observed zero. Since CNNs operate over grids, the functional embedding $E$ is evaluated at a grid of points that covers the support of all context and target points before being passed through the CNN $\rho$. Finally, kernel interpolation is used to map the discrete CNN output $\Phi(\mathcal{D})$ back to a functional one that can be queried at arbitrary locations $r(\cdot)$. In Gordon et al.'s ConvCNP, the output of the CNN has two channels that correspond to the predictive mean and variance that parameterise the (independent) Gaussian predictive at a target location.

**Transformers.** Despite frequent usage in sequence modelling tasks, the default transformer architecture (Vaswani et al., 2017) is equivariant to permutations in its inputs (Lee et al., 2019). Causal masking or positional encoding is typically used to enforce desired sequential structure, but without such augmentations and with a suitable aggregation method, the transformer is a perfectly valid set function. Lee et al. (2019) introduce an attention-based aggregation method to map from the variable-length output of a transformer to a fixed-length set of vectors. Alternatively, straightforward averaging can be used for aggregation instead, as done in the (latent path of the) attentive NP (Kim et al., 2019).

## 3. Sparse Gaussian Neural Processes

### 3.1. A Primitive Approach

When training a sparse variational Gaussian process of the kind specified in Section 2.1, our task boils down to selecting the variational parameters $\mathbf{Z}$, $\mathbf{m}$, and $\mathbf{S}$. One way to meta-learn the selection of these parameters is to amortise the inference across tasks—to perform amortised variational inference (AVI; Margossian and Blei, 2024; Ganguly et al., 2024) and train an *inference network* to map from task-specific observations to task-specific variational parameters. Since the order of the observations is irrelevant, our inference network should be a neural set function. To train such a meta-model, we optimise the parameters of the inference network with respect to the standard amortised variational inference objective function (Kingma and Welling, 2014), albeit adapted for *task*-specific rather than *sample*-specific ELBOs (Ashman et al., 2023):

$$\mathcal{L}_{AVI}(\Xi) = \frac{1}{|\Xi|} \sum_{j=1}^{|\Xi|} \mathcal{L}_{\text{ELBO}}(\mathcal{D}_c^{(j)}). \tag{16}$$

Note that there is no need for a target set, and so we take $\mathcal{D}_c = \mathcal{D}$ and $\mathcal{D}_t = \emptyset$. The naïve approach is then to prescribe a set function that maps from task observations $\mathcal{D}$ to the variational parameters $\{\mathbf{Z}, \mathbf{m}, \mathbf{S}\}$ directly. However, such an approach does little to exploit what we know about the relationships between $\mathbf{Z}$, $\mathbf{m}$, and $\mathbf{S}$; namely that the latter two depend heavily on the former. In practice, we found that such an approach either struggles to train at all (for a large meta-dataset), or overfits to near-optimal performance on tasks within the meta-dataset while catastrophically failing to generalise to new tasks (for a small meta-dataset). Hence, we propose to build a relationship-preserving inductive bias into the inference network.

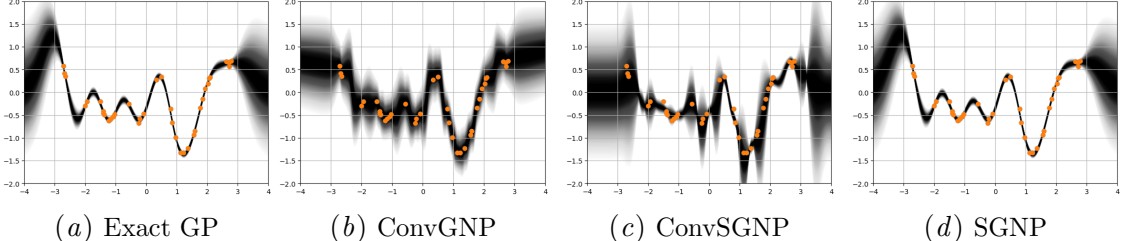

$(a)$ Exact GP $\quad\quad$ $(b)$ ConvGNP $\quad\quad$ $(c)$ ConvSGNP $\quad\quad$ $(d)$ SGNP

Figure 2: Model predictions on a synthetic regression dataset. Orange dots represent data points and the black shaded areas represent predictive distribution densities. The meta-models were trained on just five tasks.

### 3.2. Leveraging the Known Relationship Structure

In order to enforce the dependence of the inducing output parameters $\mathbf{m}$ and $\mathbf{S}$ on the inducing inputs $\mathbf{Z}$, we propose to use separate inference networks for each of the three parameter groups, and crucially, to pass $\mathbf{Z}$ (as well as the observations) into the inference networks for $\mathbf{m}$ and $\mathbf{S}$. As the inducing inputs' inference network $f$, we propose simply to use a DeepSet or transformer. Next, drawing inspiration from the Gaussian NP family (Bruinsma et al., 2021; Markou et al., 2021, 2022), we propose to use ConvDeepSets (Gordon et al., 2020) as the inference networks for $\mathbf{m}$ and $\mathbf{S}$, which we denote by $g$ and $h$ respectively. For the inducing output means $\mathbf{m}$, this is straightforward; pass the observations $\mathcal{D}_c$ through a ConvDeepSet to obtain a task-specific function-space representation $r_m(\cdot)$ and query this functional embedding at the $k$-th inducing input $\mathbf{z}_k$ to obtain the corresponding inducing output mean $m_k$

$$m_k = g(\mathcal{D}_c)(\mathbf{z}_k) \tag{17}$$
$$= r_m(\mathbf{z}_k). \tag{18}$$

For the inducing output covariance matrix inference network $h$, we adopt the kvv parameterisation as used in Markou et al. (2022):

$$S_{kl} = h(\mathcal{D}_c)(\mathbf{z}_k, \mathbf{z}_l) \tag{19}$$
$$= k\Big(h_k(\mathcal{D}_c)(\mathbf{z}_k), h_k(\mathcal{D}_c)(\mathbf{z}_l)\Big) \cdot h_v(\mathcal{D}_c)(\mathbf{z}_k) \cdot h_v(\mathcal{D}_c)(\mathbf{z}_l) \tag{20}$$
$$= k\Big(r_k(\mathbf{z}_k), r_k(\mathbf{z}_l)\Big) \cdot r_v(\mathbf{z}_k) \cdot r_v(\mathbf{z}_l) \tag{21}$$

where $k(\cdot, \cdot)$ is the exponentiated quadratic covariance function, $h_k(\cdot)(\cdot)$ is a ConvDeepSet with outputs in $\mathbb{R}^{d_k}$, and $h_v(\cdot)(\cdot)$ is a ConvDeepSet with outputs in $\mathbb{R}$ whose purpose is to modulate the covariance magnitude and allow it to shrink for inducing inputs that are closer to observations. Training is performed by optimising $\mathcal{L}_{AVI}(\Xi)$ w.r.t the parameters of all inference networks and (optionally) any hyperparameters. Note that our proposed use of $g$ and $h$ to obtain $\mathbf{m}$ and $\mathbf{S}$ is equivalent to using the Convolutional Gaussian Neural Process (ConvGNP) of Markou et al. (2022) to map from the observations and the inducing inputs to the predictive distribution over the inducing outputs, and so we refer to this model as the Convolutional Sparse Gaussian Neural Process (ConvSGNP).

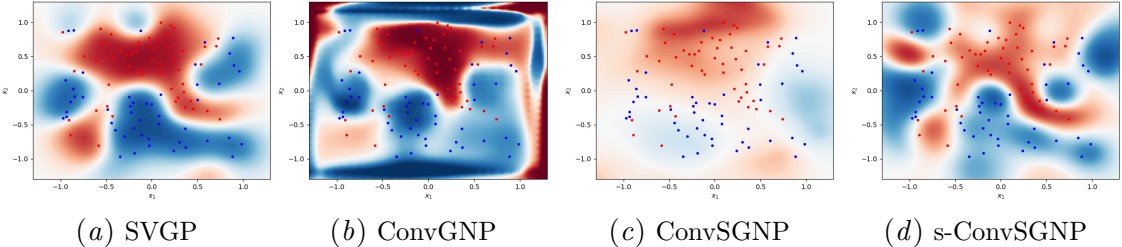

$(a)$ SVGP $\qquad$ $(b)$ ConvGNP $\qquad$ $(c)$ ConvSGNP $\qquad$ $(d)$ s-ConvSGNP

Figure 3: Model predictions on a synthetic classification dataset. Red and blue dots represent data points corresponding to opposite classes. The blue and red surface represents the predictive distribution, where each colour indicates a high probability of points belonging to that class. The meta-models were trained on just five tasks.

Aside from computational efficiency and excellent performance when used as part of a neural process (Gordon et al., 2020; Foong et al., 2020), another compelling reason to use ConvDeepSets is the fact that they are translation equivariant. Not only does this have beneficial implications regarding generalisation (Gordon et al., 2020; Ashman et al., 2024a), but since many covariance functions of practical interest are stationary, translation-equivariant inference networks allow the overall meta-model to remain translation invariant if such a covariance function is used. To enjoy the benefits of translation equivariance, however, the inducing inputs' inference network $f$ needs to be equivariant as well. What we require is for a shift $\boldsymbol{\delta} \in \mathbb{R}^d$ in the context inputs $\mathbf{X}_c^{\boldsymbol{\delta}} := \{\mathbf{x}_i + \boldsymbol{\delta}\}_{i=1}^n$ to cause an identical shift in the predicted inducing inputs $\mathbf{Z}^{\boldsymbol{\delta}} := \{\mathbf{z}_i + \boldsymbol{\delta}\}_{i=1}^m$ with no other change: $f(\mathcal{D}_c^{\boldsymbol{\delta}}) = \mathbf{Z}^{\boldsymbol{\delta}}$, where $\mathcal{D}_c^{\boldsymbol{\delta}} := \{\mathbf{X}_c^{\boldsymbol{\delta}}, \mathbf{y}_c\}$. Since the inputs and outputs of $h$ live in the same space, this is trivially achieved by subtracting the mean of the input observations before passing them into the inference network, and then adding it back to the predicted inducing inputs. In this work, we refer to a DeepSet or transformer endowed with such an augmentation as a translation equivariant (TE) DeepSet or transformer.

### 3.3. The Collapsed Shortcut

The benefits that the sparse variational GP approach of Hensman et al. (2015) provides over the Titsias (2009) approach are twofold: 1.) the ability to minibatch (Hensman et al., 2013), and 2.) the ability to use non-Gaussian likelihoods. However, in the meta-learning setting, we are not interested in minibatching since we would like to condition our predictions on *all* available observations for a particular task. This means that, as long as the likelihood is Gaussian, there is no need to meta-learn selection of the inducing output parameters $\mathbf{m}$ and $\mathbf{S}$ and we can discard the inference networks $f$, $g$, and $h$ because the Titsias equations (Eq. (6)) allow us to compute the optimal values directly. We refer to this version of the model simply as the Sparse Gaussian Neural Process (SGNP).

Although the Titsias equations are only optimal if the likelihood is Gaussian, we can still leverage the structure to our benefit in the classification setting. We propose to temporarily re-label the class labels to suitably chosen regression targets[3] (Milios et al., 2018), replace the unavailable $\sigma_y$ with a sensibly chosen $\tilde{\sigma}_y$, and then use Eq. (6) to obtain rough estimates

---

3. i.e., map class 0 to $-c$ and class 1 to $+c$ for some constant $c \in \mathbb{R}^+$

of the parameters **m** and **S**. $f$, $g$, and $h$ can then be used to refine the Titsias estimates via weighted linear interpolation. Since this model relies on the Titsias equations *as well as* the ConvDeepSets in $g$ and $h$ to find **m** and **S**, we refer to it as the semi-convolutional SGNP (s-ConvSGNP). Note that the s-ConvSGNP serves purely as a replacement for the SGNP in classification settings.

## 4. Related Work

**Gaussian Processes for Machine Learning on Related Tasks.** Gaussian processes have many desirable properties, and as a result there has been quite some interest in the problem of applying them to multiple related tasks. Bonilla et al. (2007) and Tighineanu et al. (2024) consider explicitly modelling the similarity between the current task and those observed during training to construct task-specific priors. Fortuin et al. (2020), inspired by the deep kernel learning approach of Wilson et al. (2016), instead train a neural network to approximate the best prior mean function across a meta-dataset of tasks, and similarly Patacchiola et al. (2020) perform deep kernel learning across a meta-dataset of tasks. Alternatively, Rothfuss et al. (2021, 2023) turn to the PAC-Bayes framework (McAllester, 1999) to meta-learn optimal priors for GPs. While all of these methods leverage information from related tasks to obtain a bespoke GP prior, none of them are scalable in the task size since they all require exact Gaussian process inference. By contrast, our method leverages related task information to perform rapid *and scalable* Gaussian process inference.

**Meta-Learning via Amortised Variational Inference.** Amortised variational inference provides an efficient way to perform approximate inference for many related variables. Though it is typically used to infer latent variables corresponding to samples within a dataset (Kingma and Welling, 2014), it is naturally extensible to the meta-learning setting. Liu et al. (2020) and Bitzer et al. (2023) use inference networks to map from task observations to Gaussian process hyperparameters. We go one step further by mapping to the variational parameters of a *sparse* Gaussian process, significantly improving the task-size scalability. Jazbec et al. (2021) and Ashman et al. (2020) use a sparse Gaussian process to model the latent space of a variational autoencoder for timeseries data, and use an inference network to parameterise a posterior over the inducing outputs. Contrary to their approach, we do not fix the inducing inputs to a uniform grid, and further we are interested in modelling the data space directly rather than a latent one. Ashman et al. (2023) present a conceptually similar model to ours, except they amortise inference in Bayesian neural networks rather than in sparse GPs. This means it is harder to specify meaningful priors in their model, and further, since they utilise per-data point amortisation without any sparse approximation, their approach is highly limited by the task size. Gordon et al. (2019) introduce a general framework for meta-learning approximate predictive inference via amortisation, but they maximise a predictive likelihood instead of a variational objective, meaning their approach is prone to overfitting when the number of tasks in the meta-dataset is small. Edwards and Storkey (2017) use amortised variational inference to learn meaningful fixed-length representations of datasets for downstream use, but unlike us they focus on the unsupervised setting rather than on obtaining task-specific predictions.

**Neural Processes.** We present our models as members of the NP family due to the many similarities. The ConvSGNP uses the same encoder as the ConvCNP (Gordon et al., 2020)—the ConvDeepSet—although in our model, this is only a part of the encoder, since we also use a DeepSet or transformer to parameterise $\mathbf{Z}$. Our models owe their names to the Gaussian neural process family as the ConvGNP heavily inspired their development. To highlight the differences, we contrast the ConvSGNP with the ConvGNP. While the ConvGNP maps from task observations directly to a multivariate Gaussian predictive distribution, we imbue extra structure via the inducing points. After obtaining the inducing inputs $\mathbf{Z}$ from an inference network, the ConvSGNP's treatment of the *inducing* outputs $\mathbf{u}$ is identical to the ConvGNP's treatment of the *target* outputs. We then use the sparse GP machinery to map to a predictive distribution. While the ConvGNP is free to learn any implicit prior from the data, we are interested in manually specifying a prior for the added interpretability and trustworthiness this offers to practitioners. By contrast, the SGNP is quite different to the ConvGNP—in this model, the distribution over the inducing outputs is computed in closed form rather than being parameterised by ConvDeepSets. If a transformer is used to predict $\mathbf{Z}$ (rather than a DeepSet) in any of our models, then they fall into the transformer NP family (Kim et al., 2019; Nguyen and Grover, 2022; Ashman et al., 2024c).

## 5. Experiments

**Synthetic 1D Regression.** We begin the evaluation of our model with a one-dimensional regression problem on data sampled from a GP prior with a squared-exponential (SE) covariance function. We compare the ConvSGNP and SGNP with Markou et al. (2022)'s ConvGNP, and we consider meta-datasets of size $|\Xi| = 1000$ and $|\Xi| = 5$. We also compare to a GP with the data-generating hyperparameters evaluated seperately on each test task's context data. The ConvSGNP and SGNP used 32 inducing points and a transformer as $f$. We visualise the model predictives on a test task for the $|\Xi| = 5$ case in Fig. 2, and we summarise the quantitative results in Table 1. Although the ConvSGNP exhibits poor performance, the SGNP outperforms the ConvGNP and closely rivals the oracle model in both settings. Unlike the SGNP, the performance of the ConvGNP is significantly degraded when the number of observed tasks is limited.

| $|\Xi|$ | GP | ConvGNP | ConvSGNP | SGNP |
|---|---|---|---|---|
| 5 | $0.10 \pm 0.06$ | $-2.02 \pm 0.44$ | $-4.33 \pm 0.76$ | $\mathbf{0.06} \pm 0.07$ |
| 1000 | | $-0.23 \pm 0.06$ | $-1.57 \pm 0.22$ | $\mathbf{0.10} \pm 0.06$ |

Table 1: Average per-target-datapoint log-likelihoods (higher is better) achieved by each model across 100 test tasks.

**Synthetic 2D Classification.** Next, we consider a binary classification problem in which data is sampled from a Bernoulli distribution with logit space given by a 2D, SE covariance, GP prior sample. We replace the SGNP with the s-ConvSGNP, and we use a sparse variational GP (SVGP; Hensman et al., 2015) as the oracle since exact GP classification is intractable. The models with inducing points used 64, and the SGNP-based models used a transformer for $f$. To adapt the ConvGNP for classification, we push the Gaussian pre-

dictive through the standard Gaussian CDF to compute a Bernoulli predictive distribution (Bishop, 2006). Once again, we consider meta-datasets of size $|\Xi| = 1000$ and $|\Xi| = 5$ with model predictives for an unseen task for the $|\Xi| = 5$ case shown in Fig. 3, and the quantitative results are summarised in Table 2. While the ConvGNP and s-ConvSGNP both achieve near oracle-level performance in the data-rich setting, the ConvGNP's performance suffers much more in the data-sparse setting. The ConvSGNP also exhibits less degradation than the ConvGNP, with superior performance to it when the meta-dataset is small.

| $\lvert\Xi\rvert$ | SVGP | ConvGNP | ConvSGNP | s-ConvSGNP |
|---:|:---:|:---:|:---:|:---:|
| 5 | $-0.56 \pm 0.01$ | $-1.07 \pm 0.03$ | $-1.01 \pm 0.09$ | $\mathbf{-0.79} \pm 0.03$ |
| 1000 | | $\mathbf{-0.57} \pm 0.01$ | $-0.68 \pm 0.01$ | $\mathbf{-0.58} \pm 0.01$ |

Table 2: Average per-data point log-likelihoods (higher is better) achieved by each model on unseen data across 100 test tasks.

**Tétouan City Power Consumption.** Finally, we consider the challenging real-world task of modelling the power consumption in the Moroccan city of Tétouan (Salam and El Hibaoui, 2018). Power consumption data is available for three different zones within the city and has been collected at ten minute intervals over the course of 2017. We imagine it is the end of February and that we have had access to the zone 1 and 2 data for some time (i.e. enough time to train a meta-learner), and that we have recently received some data corresponding to city zone 3. We consider two contrasting prediction scenarios. Firstly, we suppose we have received an incomplete dataset for January and February in zone 3 and that we need to predict the missing values. Secondly, we suppose we have received the full dataset but that forecasts are needed for the upcoming March in zone 3. We refer to these as the *interpolation* and *extrapolation* problems respectively.

We use an SGNP with 256 inducing points and a TE DeepSet for $f$. We consider a conditional NP (CNP), a ConvCNP, a ConvGNP, as well as diagonal (TNP-D) and non-diagonal (TNP-ND) transformer neural processes Nguyen and Grover (2022) as baselines. We expect power consumption to follow daily and weekly trends, so we use a bespoke covariance function in the SGNP that consists of the sum of two periodic kernels with fixed daily and weekly periodicities (that only act on the time feature) and an SE kernel with ARD (that acts on all features). Such prior elicitation is not possible in any of the baselines. Furthermore, we consider a Titsias (2009)-style sparse GP regressor (SPGR) with 256 inducing points and the same composite covariance function, which is trained on each test task's context data directly. To highlight the impracticality of using a sparse GP, we include prediction times for each model in our results, which are in Table 3. The SGNP significantly outperforms all baseline NPs and either rivals or outperforms the SGPR. The SGNP also rivals the CNP in terms of prediction speed.

## 6. Discussion

**Our Work.** We have presented a new type of model that can be equivalently interpreted as a neural process that is amenable to GP-prior elicitation, or a meta-learner that rapidly implements sparse variational GP inference at test-time. Our model is much more *com-*

| Problem | Metric | SPGR | CNP | ConvCNP | ConvGNP | TNP-D | TNP-ND | SGNP |
|---|---|---|---|---|---|---|---|---|
| Interp. | LL ($\uparrow$) | $1.13 \pm 0.03$ | $-22.30 \pm 1.10$ | $-3.60 \pm 0.15$ | $0.74 \pm 0.00$ | $-44.31 \pm 1.43$ | $-13.22 \pm 0.00$ | $\mathbf{0.92 \pm 0.01}$ |
| | MAE ($\downarrow$) | $0.44 \pm 0.01$ | $3.07 \pm 0.04$ | $1.20 \pm 0.01$ | $1.68 \pm 0.02$ | $2.35 \pm 0.03$ | $2.90 \pm 0.03$ | $\mathbf{0.47 \pm 0.01}$ |
| | time [s] ($\downarrow$) | $1609.30$ | $0.01$ | $2.71$ | $74.11$ | $1.62$ | $1.81$ | $0.06$ |
| Extrap. | LL ($\uparrow$) | $-0.05 \pm 0.02$ | $-2574.08 \pm 81.36$ | $-1.36 \pm 0.14$ | $-100.14 \pm 0.00$ | $-154.32 \pm 4.57$ | $0.13 \pm 0.00$ | $\mathbf{0.25 \pm 0.02}$ |
| | MAE ($\downarrow$) | $1.48 \pm 0.02$ | $17.16 \pm 0.28$ | $6.93 \pm 0.05$ | $5.28 \pm 0.03$ | $3.73 \pm 0.04$ | $3.73 \pm 0.04$ | $\mathbf{1.12 \pm 0.02}$ |
| | time [s] ($\downarrow$) | $3650.23$ | $0.01$ | $58.03$ | $275.96$ | $3.86$ | $4.03$ | $0.10$ |

Table 3: Average per-(target-)datapoint log-likelihood (LL) and predictive mean absolute error (MAE, in megawatts) achieved in the Tétouan city power consumption experiment. Prediction time is measured from arrival of context data to completion of predictions. We embolden the LL and MAE of the best performing meta-learner.

*putationally* efficient than existing GP-based meta-learners, yet much more *data* efficient and interpretable than existing neural processes. This means it is particularly well suited to problems in which some combination of the following properties hold: 1. the number of datapoints in each task is beyond the GP-feasible limit, 2. domain knowledge is available in GP-prior form, 3. model interpretability is desirable, and 4. the number of observed tasks is small. Our synthetic-data experiments highlighted such utility with the fact that the (s-Conv)SGNP performance remains strong when a very small meta-dataset ($\Xi = 5$) is used, while the ConvGNP performance degrades significantly. We attribute the poorer performance of the ConvSGNPs relative to the (s-Conv)SGNPs to known difficulties with Hensman et al.-style SVGP loss-landscapes (Bauer et al., 2016). In the Tétouan city power consumption experiment where all four of the above properties hold, the SGNP is by far the most performant meta-learner and the only one which allows us to imbue our prior knowledge. Furthermore, the SGNP outperforms the SPGR in the extrapolation scenario—we suspect this is because meta-learning kernel parameters across tasks allows it to learn hyperparameters than generalise better. Further discussion can be found in Appendix A.

**Future Work.** There are a number of interesting questions regarding Sparse Gaussian Neural Processes that provide fertile ground for future research. ***Hyperparameter Selection.*** We assumed that hyperparameters are shared across tasks within a problem setting. However, if different tasks were to correspond to different hyperparameters, then in principle our models can be augmented with an additional neural set function to map from task observations to task-specific hyperparameters. ***Model Misspecification.*** The performance of GPs can be poor if the model is misspecified (Rasmussen and Williams, 2005). However, we suspect that our models might be able to defend against poor predictions due to model misspecification via a different objective function. Instead of optimising the ELBO, which would encourage similar behaviour to the true but misspecified GP posterior, we can borrow a neural process objective function that encourages good predictions on target points. Such objective functions have been shown to recover the mapping to the ground-truth stochastic process for suitably flexible models (Foong et al., 2020). ***Extension to Other Variational Models.*** In principle, one could use neural set functions to map to the approximate posterior of *any* variational model via per-dataset amortised inference. This means that our models' performance could potentially be improved by using better sparse variational GP approximations (Shi et al., 2020; Bui et al., 2025; Titsias, 2025). Alternatively, one could go beyond GPs and meta-learn sparse Bayesian neural network inference via, for example, Ober and Aitchison (2021)'s method.

## Acknowledgments

VF was supported by the Branco Weiss Fellowship.

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

## Appendix A. Further Discussion

**Validity as a Neural Process.** Our model can be viewed as a member of the *conditional* NP family. A conditional NP encodes a context set into a deterministic representation, and then decodes the combination of the representation and the test points into a probabilistic prediction. In our model, the inducing inputs $\mathbf{Z}$ and the inducing output parameters $\mathbf{m}$ and $\mathbf{S}$ serve as the context set representation. Using the Bayesian machinery for conditioning Gaussian processes, the test points are combined with the representation to obtain posterior predictive distributions. The inference networks then serve as the neural process encoder, and we have a nonparametric decoder. Passing the latent variable posterior parameters directly to the decoder instead of a latent variable sample is also what Volpp et al. (2021) do to convert a latent variable NP to a conditional one.

Alternatively, our model can be interpreted as a neural process with both a conditional and latent path, somewhat akin to in the original ANP (Kim et al., 2019). The inducing inputs $\mathbf{Z}$ serve as the representation in the conditional path since they are deterministic in nature, while the inducing outputs $\mathbf{u}$ serve as the probabilistic latent variable. The key innovation of our approach is to place a GP prior over the latent variable (rather than the typical spherical Gaussian). Inference over the latent variable is then performed either in closed form via the Titsias equations or in amortised fashion via a ConvDeepSet. An important difference between the SGNP's latent path and a traditional NP's latent path is that the predictive distribution is obtained by analytic marginalisation of the latent variable. In the non-conjugate case of the ConvSGNP, Monte Carlo sampling of the latent variable is required as normal to approximate the predictive density.

**Computational Complexity.** In Table 4 we outline the computational complexity associated with a training-stage forward pass through the various neural set functions, models, and meta-models considered in or relevant to this paper. Note that the number of gridpoints $t$ tends to scale exponentially in the number of data dimensions, meaning that while ConvDeepSets technically have linear time complexity, $t$ itself can quickly become prohibitively large.

**Model Properties.** In Table 5 we highlight which properties are exhibited by our model (SGNP) in comparison with GPs, sparse GPs (SGPR and SVGP), and the rest of the NP family. Further explanations of what we mean by each property decsription are as follows:

1. *Prior elicitation*: can we manually elicit prior beliefs (i.e. without training on synthetic/prior data)?

2. *Task-efficient*: does the method avoid the need for copious training tasks?

3. *Scalable*: is the method applicable for tasks with many observations?

4. *Fast adaptation*: does the method avoid the need for optimisation loops to make predictions on new tasks?

5. *Interpretable*: are the method's predictions interpretable/explainable?

6. *Modelling flexibility*: can the method recover arbitrarily complex stochastic process posteriors, such as non-Gaussian posteriors or those with non-closed-form covariance

| DeepSet | ConvDeepSet | Transformer |
|---|---|---|
| $\mathcal{O}(n)$ | $\mathcal{O}(n+t)$ | $\mathcal{O}(n^2)$ |

| GP | SGPR | SVGP |
|---|---|---|
| $\mathcal{O}(n^3)$ | $\mathcal{O}(nm^2)$ | $\mathcal{O}(bm^2 + m^3)$ |

| NP | ANP/TNP | PT-TNP |
|---|---|---|
| $\mathcal{O}(n_c + n_t)$ | $\mathcal{O}(n_c^2 + n_c n_t)$ | $\mathcal{O}(mn_c + mn_t)$ |

| ConvGNP | ConvSGNP | SGNP |
|---|---|---|
| $\mathcal{O}(n_c + t)$ | $\mathcal{O}(t + n_c m^2 + n_c^2)$ | $\mathcal{O}(n_c m^2 + n_c^2)$ |

Table 4: $n$ is the number of datapoints, $t$ is the number of grid points, $b$ is the number of points in a minibatch, $m$ is the number of inducing points or pseudo-tokens (in the PT-TNP). The $+n_c^2$ (red) terms are only present if a transformer is used as the inducing input inference network $f$ in the SGNP models. We assume that $n_c > m$. Best viewed in colour.

structure? For example, consider the classic NP task of pixelwise regression over facial images—NPs can learn an implicit "face kernel", but GPs are restricted to whatever closed-form kernel we choose (e.g. SE, weakly periodic etc).

This table highlights the fact that our method essentially swaps the ability to model arbitrarily complex stochastic processes with the ability to manually select the stochastic process family. Perhaps the main takeaway of our paper is that total flexibility leads to overfitting when we have limited tasks, whereas the ability to guide an NP via prior elicitation leads to greater task-efficiency.

| Property | GP | Sparse GP | NP family | SGNP |
|---|---|---|---|---|
| Prior elicitation | ✓ | ✓ | ✗ | ✓ |
| Task-efficient | ✓ | ✓ | ✗ | ✓ |
| Scalable | ✗ | ✓ | ✓ | ✓ |
| Fast adaptation | ✓ | ✗ | ✓ | ✓ |
| Interpretable | ✓ | ✓ | ✗ | ✓ |
| Modelling flexibility | ✗ | ✗ | ✓ | ✗ |

Table 5: Properties of the various types of model considered in this work.

## Appendix B. Experimental Details

The code for this project can be found at https://github.com/Sheev13/sgnp.

### B.1. Synthetic 1D Regression

**Data Generating Process.** Each task in the meta-dataset was constructed in the following way. We uniformly sample an integer $n$ from the range $[10, 100]$, and then sample $n$ points uniformly from the range $[-3, 3]$ which serve as the inputs $\mathbf{X}$. Next, a function is sampled from a zero-mean Gaussian process prior with covariance function given by

$$k(x_1, x_2) = \exp\left(-\frac{(x_1 - x_2)^2}{2l^2}\right) \tag{22}$$

where the lengthscale parameter is set to $l = 0.5$, and the function is evaluated at the $n$ inputs. Independent observation noise of variance $\sigma_y^2 = 0.05^2$ is added to each function value to obtain the noisy labels $\mathbf{y}$.

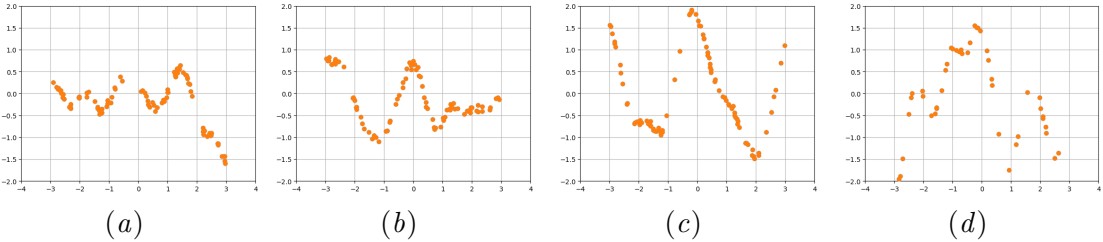

Figure 4: Four example datasets generated in the same way as those used in the meta-dataset used in the synthetic 1D regression experiment.

The test task data were generated the same way, save that $n$ was sampled uniformly from the range $[5, 30]$. Furthermore, a random proportion $p$ of the points in each task were selected as context points with the remaining points used as targets for evaluation. $p$ was uniformly sampled from the range $[0.45, 0.55]$ for each task.

**Architectures.** The ConvGNP used ConvDeepSets in which the CNNs had three convolutional layers with 32 channels, a kernel of size 5, and ReLU nonlinearities. The ConvDeepSets used a grid resolution of 50 points per unit (2e-2 spacing), and the kvv block used $d_k = 8$. All ConvDeepSet kernel lengthscales were freely optimised during training.

The SGNP, with $m = 32$ inducing variables, used a translation-equivariant transformer as the inference network for the inducing inputs. The translation-equivariance was achieved by pre- and post-processing of the data as specified in Section 3.2. The transformer itself consisted of two sequential multi-head self-attention blocks each implemented as a `torch.nn.TransformerEncoderLayer`. 8 heads were used, feedforward layers were of width 32, the tokens were of dimension 32, there was no dropout, and ReLU nonlinearities were used throughout. (Pre-processed) input-output pairs were concatenated and passed through a learnable linear transformation to obtain initial 32-dimensional tokens. For a context set of cardinality $n_c$, the output of the transformer is a tensor in $\mathbb{R}^{n_c \times 32}$. This tensor is averaged over the batch (first) dimension to obtain a vector in $\mathbb{R}^{32}$, which is then passed

through a learnable linear transform to $\mathbb{R}^{m \cdot d} = \mathbb{R}^{32}$, which is then reshaped into a stack of 32 vectors in $\mathbb{R}^d = \mathbb{R}$ which are the inducing inputs (before post-processing). A squared-exponential kernel with learnable input lengthscale and signal output scale, as well as a Gaussian likelihood with learnable observation noise were used.

The ConvSGNP had the same setup at the SGNP, but with the Titsias equations replaced by a ConvGNP of the same architecture and setup as the ConvGNP baseline.

**Training.** All meta-learners used the same meta-dataset in each case of meta-dataset size. 20,000 training steps were performed, with the average loss over a minibatch of 5 tasks used to inform each training step's direction. All meta-learners used a learning rate that was linearly scheduled from 1e-3 down to 5e-5. The SGNP and ConvSGNP used the variational objective defined in Eq. (16), while the ConvGNP used a maximum predictive likelihood objective defined on a target set of points. For the ConvGNP, the proportion of points used as context points was uniformly sampled from [0.25, 0.75], while all points were used as targets.

### B.2. Synthetic 2D Classification

**Data Generating Process.** Each task in the meta-dataset was constructed in the following way. We uniformly sample an integer $n$ from the range $[30, 150]$, and then sample $n$ points uniformly from the range $[-1, 1] \times [-1, 1]$ which serve as the inputs $\mathbf{X}$. Next, a function is sampled from a zero-mean Gaussian process prior with covariance function given by

$$k(x_1, x_2) = 2.5 \exp\left(-\frac{|\mathbf{x}_1 - \mathbf{x}_2|^2}{2l^2}\right) \tag{23}$$

where the lengthscale parameter is set to $l = 0.25$, and the function is evaluated at the $n$ inputs. The function values then serve as the logits for $n$ independent Bernoulli distributions, each of which is sampled to obtain the $n$ binary labels $\mathbf{y}$.

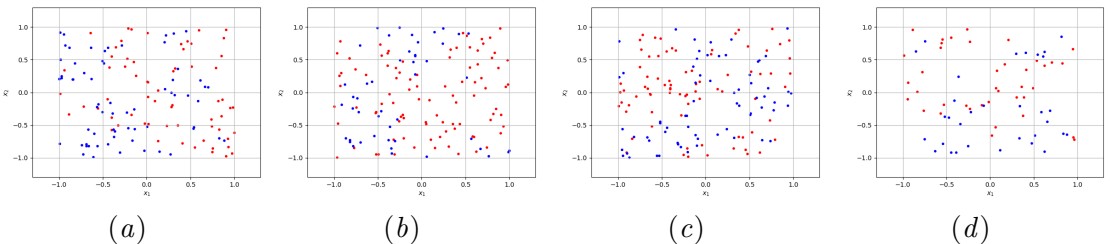

Figure 5: Four example datasets generated in the same way as those used in the meta-dataset used in the synthetic 2D classification experiment.

The test task data were generated the same way, save that $n$ was sampled uniformly from the range $[60, 300]$. Furthermore, a random proportion $p$ of the points in each task were selected as context points with the remaining points used as targets for evaluation. $p$ was uniformly sampled from the range $[0.45, 0.55]$ for each task.

**Architectures.** The ConvGNP used ConvDeepSets in which the CNNs had two convolutional layers with 32 channels, a kernel of size 3, and ReLU nonlinearities. The Con-

vDeepSets used a grid resolution of 10 points per unit (1e-1 spacing) in both input dimensions, and the kvv block used $d_k = 8$. All ConvDeepSet kernel lengthscales (including seperate ones for each input dimension) were freely optimised during training.

The s-ConvSGNP, with $m = 64$ inducing variables, used a translation-equivariant transformer as the inference network for the inducing inputs. This transformer had identical setup to that of the SGNP in the 1D regression experiment, save that the input and output linear transformations were modified to handle 2D inputs and 64 inducing variables. The ConvDeepSets were set up identically to those in the ConvGNP benchmark. For the Titsias equations, the binary labels were mapped to regression labels of $\pm 2.0$ and a pseudo observation noise of $\tilde{\sigma}_y = 0.5$ was used. The inducing output mean vector and covariance matrix were obtained by linearly interpolating between the Titsias-estimated parameters and ConvDeepSet-estimated parameters, where the point $1/10$ along the interpolation line was chosen in both cases (i.e. the Titsias estimates are favoured). A squared-exponential kernel with learnable input lengthscales for each input dimension and signal output scale, as well as a Gaussian likelihood with learnable observation noise were used.

The ConvSGNP had the same setup at the s-ConvSGNP, just without any use of Titsias-estimates.

**Training.** All meta-learners used the same meta-dataset in each case of meta-dataset size. 10,000 training steps were performed, with the average loss over a minibatch of 5 tasks used to inform each training step's direction. All meta-learners used a learning rate that was linearly scheduled from 1e-3 down to 5e-5. The s-ConvSGNP and ConvSGNP used the variational objective defined in Eq. (16), while the ConvGNP used a maximum predictive likelihood objective defined on a target set of points. For the ConvGNP, the proportion of points used as context points was uniformly sampled from [0.05, 0.5], while all points were used as targets.

### B.3. Data Pooling

During peer-review of this work, the question came up of why we cannot just train a (SV)GP on a dataset constructed by pooling those in the meta-dataset. In short, the answer is because we assume the tasks in the meta-dataset are different enough that pooling them results in gibberish.

Let's consider the case for 1D GP-generated data. Consider sampling many functions from a GP prior and keeping finitely many randomly chosen and noisy input-output pairs from each function. The resulting (single) dataset will appear increasingly similar to one sampled from a white noise process as the number of sampled GP functions increases. That is, as the number of pooled datasets increases, any underlying squared-exponential covariance structure will become harder and harder to distinguish. We do not provide a formal argument for this, but we provide some examples that demonstrate this in Fig. 6. What this means is that any (SV)GP evaluated on such a dataset will not be capable of meaningful predictions on any particular task within the pooled dataset.

### B.4. Tétouan City Power Consumption

**Dataset Curation.** Due to limited computational resources, we restricted ourselves to using only the first three months of the Salam and El Hibaoui (2018) dataset. This was

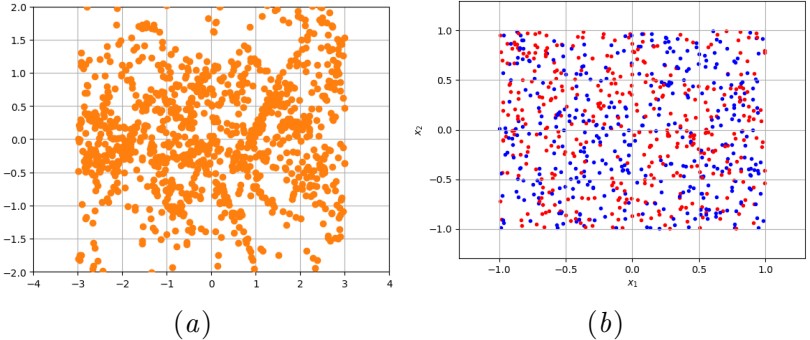

Figure 6: Examples of datasets created by pooling multiple datasets in a meta-dataset.

especially important for some of the baselines, particularly the ConvGNP, rather than our model. Since the convolutional models (ConvCNP, ConvGNP) are only applicable to data with up to three dimensions, we only use the time, temperature, and humidity features of the dataset for all models. This is because convolutional layers of 4 or more dimensions are not generally available. The SGNP does not share this limitation. The input data for the three city zones are the same, and so they are normalised to have zero mean and unit standard deviation in each feature dimension. The targets for zone 1 and 2 are concatenated and the mean and standard deviation of this augmented set is used to normalise each zone's target data (including zone 3). For the interpolation task, a random 50/50 split is performed on the zone 3 data to obtain the context and target set. This split is the same for all models. For the extrapolation task, all January and February zone 3 data are used as contexts and all March zone 3 observations are used as targets. In both cases, we therefore have a meta-dataset of size $|\Xi| = 2$ where the two datasets correspond to each of city zones 1 or 2 and each contain $(31 + 28) \times 24 \times 6 = 8496$ datapoints. Exact Gaussian process regression was therefore not possible (at least with our limited computing resources).

**Architectures.** The CNP used three fully-connected layers of 128 units in the encoder, an embedding dimension of 128, and three fully-connected layers of 128 units in the decoder. ReLU nonlinearities were used throughout.

The ConvCNP used a ConvDeepSet in which the CNN had three convolutional layers with 128 channels, a kernel size of 3, and ReLU nonlinearities. The ConvDeepSet used a grid spacing of 1.25e-2 for the time feature and 0.5 for each of the temperature and humidity features. These numbers were chosen to be small enough to be able to capture any variation, but as large as possible to minimise the computational load. Note that they correspond to post-normalisation feature scales. All ConvDeepSet kernel lengthscales were freely optimised during training, but we found that stability of training was highly sensitive to lengthscale initialisation. To overcome this, we used very small initialisations (one tenth of the corresponding grid spacing for a particular feature) and trained a ConvCNP under sigmoid nonlinearities, whose training we found to be much more robust to poor lengthscale initialisations. The final learned lengthscales were then used as the initial lengthscales for the ReLU-nonlinearity ConvCNP. Although all datapoints lie on a grid (of ten minute intervals) in the time-dimension, they do not in the other dimensions, and so we cannot use the on-the-grid variant of the ConvCNP.

The ConvGNP used ConvDeepSets with identical setups to the one in the ConvCNP. The `kvv` block used $d_k = 16$. We intended to perform the same trick of training a ConvGNP with sigmoid nonlinearities to find a good initial set of lengthscales for training a ReLU nonlinearity ConvGNP. However, the sigmoid ConvGNP took a week to train and we found that the sigmoid and ReLU ConvCNPs that we had trained performed almost identically to each other. Therefore, we decided to avoid an extra week of computation and just use the trained sigmoid ConvGNP for the experiment.

The TNP-D used three sequential multi-head self-attention blocks (again implemented via a `torch.nn.TransformerEncoderLayer`) each with 8 heads, feedforward layer widths of 128, embedding dimensions of 128, no dropout, and ReLU nonlinearities. Furthermore, the final target set tokens were decoded to predictive means and variances via a ReLU-nonlinearity MLP with two layers of 128 dimensions.

The TNP-ND had identical architecture to the TNP-D, save that the final MLP only mapped to the predictive mean. The covariance matrix over the target outputs was constructed by passing the final target set tokens through another self attention block (also implemented as a `torch.nn.TransformerEncoderLayer` with 128 dimensions everywhere and ReLU nonlinearities), and then passing these through a ReLU MLP with two layers of 128 units to obtain outputs of dimension $d_k = 16$. The output $n_t \times d_k$ tensor was then matrix multiplied with a transposed version of itself, and the lower triangular portion of the result was used as the $n_t \times n_t$ lower triangular Cholesky factor of the covariance matrix.

The SGNP had $m = 256$ inducing variables, and used a translation-equivariant DeepSet as the inference network for the inducing inputs. The translation-equivariance was achieved by pre- and post-processing of the data as specified in Section 3.2. The DeepSet consisted of three fully connected layers with 128 units and ReLU nonlinearities. The likelihood was Gaussian with learnable observation noise. The covariance function consisted of the sum of two periodic kernels and an ARD kernel. The periodic kernels were of the form

$$k_{per}(t, t') = \sigma_f^2 \exp\left(\frac{-2\sin\left(\pi\frac{t-t'}{p}\right)}{l^2}\right) \tag{24}$$

where the period $p$ was fixed to correspond to the length of a day and week in each, the output scale $\sigma_f$ and lengthscale $l$ were freely optimised, and they only acted on the time feature. The ARD kernel was of the form

$$k_{ARD}(\mathbf{x}, \mathbf{x}') = \sigma_f^2 \exp\left(-\frac{1}{2}\left|\frac{\mathbf{x} - \mathbf{x}'}{\mathbf{l}}\right|^2\right) \tag{25}$$

where the lengthscale vector is of the same dimensionality as the inputs (so the division is elementwise), and output scale and all lengthscales were freely optimised.

The SPGR used the same composite kernel as described above, had 256 inducing points, and also used a Gaussian likelihood with learnable observation noise.

**Training.** All models were trained for $10,000$ steps with Adam under default torch settings, with an initial learning rate of 1e-3 linearly tempered down to 5e-5. All steps were full-batch, meaning the loss was computed using both tasks (zone 1 and zone 2) for the meta-learners before updating the parameters, and using all context points for the SPGR

before updating the parameters. The SGNP and SGPR both used the Hensman et al. (2015)-style ELBO defined in Eq. (16) with 5 posterior samples to estimate the expected log-likelihood term. The baseline neural processes were trained via maximum predictive likelihood. The proportion of points used as context points was uniformly sampled from [0.2, 0.8] at each training step, while all points were used as targets. However, due to limited compute, the ConvCNP first selected a random 20% of points as the "full set" in each training iteration, while the ConvGNP did the same with a random 15%, and the TNPs (-D and -ND) with a random 25%. We reproduce Table 3 here along with a third row of results corresponding to an interpolation task on the January and February data for zone 1. This is one of the tasks that the meta-learners were trained on (though not necessarily with this *particular* 50/50 context-target split), and so we use this to demonstrate how the baselines perform well in training but fail to generalise to new tasks. For the SGPR, this task is analogous to the zone 3 interpolation task, but its results are included for completeness. Each meta-learner was trained just once (again due to limited compute), and the uncertainty reflects the standard deviation of the sample mean (across target points) of each metric. For models with multivariate likelihoods (ConvGNP and TNP-ND), this means that there was no variation in per-datapoint log likelihoods since the target set log likelihood was just divided by the number of target points.

| Problem | Metric | SPGR | CNP | ConvCNP | ConvGNP | TNP-D | TNP-ND | SGNP |
|---------|--------|------|-----|---------|---------|-------|--------|------|
| Interp. | LL ($\uparrow$) | $1.13 \pm 0.03$ | $-22.30 \pm 1.10$ | $-3.60 \pm 0.15$ | $0.74 \pm 0.00$ | $-44.31 \pm 1.43$ | $-13.22 \pm 0.00$ | $0.92 \pm 0.01$ |
|         | MAE ($\downarrow$) | $0.44 \pm 0.01$ | $3.07 \pm 0.04$ | $1.20 \pm 0.01$ | $1.68 \pm 0.02$ | $2.35 \pm 0.03$ | $2.90 \pm 0.03$ | $0.47 \pm 0.01$ |
| Extrap. | LL ($\uparrow$) | $-0.05 \pm 0.02$ | $-2574.08 \pm 81.36$ | $-1.36 \pm 0.14$ | $-100.14 \pm 0.00$ | $-154.32 \pm 4.57$ | $0.13 \pm 0.00$ | $0.25 \pm 0.02$ |
|         | MAE ($\downarrow$) | $1.48 \pm 0.02$ | $17.16 \pm 0.28$ | $6.93 \pm 0.05$ | $5.28 \pm 0.03$ | $3.73 \pm 0.04$ | $3.73 \pm 0.04$ | $1.12 \pm 0.02$ |
| Zone 1  | LL ($\uparrow$) | $0.67 \pm 0.02$ | $-0.04 \pm 0.02$ | $0.56 \pm 0.03$ | $1.42 \pm 0.00$ | $1.05 \pm 0.02$ | $0.69 \pm 0.00$ | $0.40 \pm 0.03$ |
|         | MAE ($\downarrow$) | $0.74 \pm 0.01$ | $3.04 \pm 0.05$ | $0.85 \pm 0.01$ | $1.85 \pm 0.03$ | $1.23 \pm 0.03$ | $3.48 \pm 0.05$ | $0.92 \pm 0.01$ |

Table 6: Average per-(target-)datapoint log-likelihood (LL) and predictive mean absolute error (MAE, in megawatts) achieved in the Tétouan city power consumption experiment, as well as on a task representing what the meta-models saw during training.

