# OpenReview forum: "Sparse Gaussian Neural Processes"
_approximateinference.org/AABI/2025/Proceedings_Track — AABI 2025 Proceedings Track_

### Official Review · Reviewer_D9rJ · 2025-02-21
**Sparse Gaussian Neural Processes: Bridging Interpretability, Efficiency, and Scalability in Meta-Learning**

**Rating:** 7
**Confidence:** 4

**Review:**

The paper presents a technically rigorous and well-motivated study on integrating sparse variational Gaussian process (GP) inference with neural processes (NPs). The work is methodologically solid and mathematically detailed. Key elements include:

Mathematical Rigor:
The authors provide clear derivations and formulae; this level of detail demonstrates strong technical depth.

Experimental Validation:
Extensive experiments are reported on both synthetic and real-world datasets, such as the Tétouan city power consumption case. The quantitative results are summarized in tables, and visualizations (Figs. 2–3) help demonstrate the benefits of the proposed approach.

Expository Style:
The paper systematically introduces background material on sparse variational GPs and neural processes before presenting the new model. This layered approach helps readers build up the necessary context. Figures comparing models (e.g., ConvGNP vs. SGNP) and tables of log-likelihoods add to the clarity of the experimental results, though some sections may require familiarity with GP and NP literature.

Originality
The work stands out by combining ideas from both sparse Gaussian processes and neural processes in a novel way. The introduction of the Sparse Gaussian Neural Process (SGNP) represents a creative synthesis—leveraging the interpretability and prior elicitation of GPs while retaining the meta-learning benefits of NPs. By enforcing known relationship structures via network design, the work enhances interpretability—a feature often missing in many deep models.

Practical Significance in Low-Data Regimes:
The model is particularly tailored for settings with a small number of tasks or when prior domain knowledge is available, a scenario common in high-stakes applications where uncertainty estimates are critical.

Significance
The proposed approach has the potential to make a notable impact in probabilistic meta-learning and GP-based methods:

Bridging Two Paradigms:
By uniting sparse variational GP techniques with neural processes, the paper provides a pathway to fast, interpretable predictions while leveraging meta-learned priors. This can be especially useful in scenarios where rapid adaptation to new tasks is essential.
Interpretability in Deep Learning:
The emphasis on clear, interpretable priors is significant given the ongoing demand for explainable machine learning, particularly in high-stakes decision-making.
Experimental Insights:
The reported performance improvements—especially in low-task regimes—suggest that the SGNP and its variants (like the s-ConvSGNP for classification) may offer practical benefits over more flexible but less interpretable models.

---

### Official Review · Reviewer_iZen · 2025-02-28

**Rating:** 5
**Confidence:** 4

**Review:**

Post-response update. The authors gave an insightful response, which clarified my concerns. I think this work would be interesting for AABI community. I'm raising my score to borderline 5.

-----

The paper proposes an SVGP model where the variational parameters $m,S$ of the inducing posterior $q(u) \sim N(m,S)$ are the output of a meta neural network of various datasets. The model adds dataset amortisation, which is a flexible and efficient way to infer the variational parameters. The idea is clever, useful and novel in this context.

The experiments are small, and only compare to the immediate competing meta network approaches. In GP literature the standard is to perform wide comparisons to many other GP methods, and over a wide range of datasets. The experiments are anecdotal, and not convincing. The experiments also do not clearly show evidence for the paper's claims. The paper claims its method to be interpretable and offer "prior elucidation", and solve the problem of large datasets. I don't see empirical evidence for these.

The paper is generally well-written, but crucially the setting is confusing. It is not clear what the paper explicitly means by tasks or meta-learning or multiple datasets. For instance, why can't we just pool the data? The model is described only as modifications over earlier models, which makes the model description incomplete and vague. I also did not understand what even is being learnt in this paper since the losses do not define what parameters are being optimised. None of the visualisations show the tasks or datasets or parameters.

The paper proposes a really good idea and promising results, but due to the insufficient experiments and insufficient presentational clarity I don't think this paper should be published yet. With a bit of extra work, this could become a strong paper. In it's current form I then rate this as reject

Below is more specific points.

- z is used for both task variables and inducing points: I would avoid this.
- Eqs 5 and 16 are vague wrt what parameters are being optimised. Please define
- “Task” is undefined and not clear. Similarly it's unclear how the different datasets differ from each other.
- What’s the difference between D^(j) and D_c?
- In eq 18 the g(D_c) turns into r_k, which does not depend on _c anymore: why?
- How is the g(D)(z) computed in practise?
- Secs 2.1 and 2.2 give careful interpretations of likelihoods, posteriors and ELBOs. However, sec 3 omits all of this. The apper should make these interpretations explicit wrt the sec 3 model, and also give the sec 3 model in a complete, explicit and self-contained way.
- Does |Xi|=1000 mean that you use 1000 datasets? Is this a realistic assumption in the intended usecases for this model? How many datapoints are in each dataset?
- What are the different tasks in synthetic experiments, and what is the “meta” aspect here?
- Figs 2+3 only visualise the final predictives, while it should also visualise the tasks, datasets, ground-truth, and parameters.
- What is oracle model?
- How is the standard GP fitted with multiple datasets?
- Power experiment mentions “context data”. What is this in the sec 3 model?
- Can you provide experiments against a baseline of pooling the datasets
- How is the (SV)GP trained with multiple datasets?
- In the experiments the (SV)GP is always the best. Why not then use that? If the paper argues for scalability, it should include memory/runtime analysis.
- The discussion claims that the model is highly interpretable and well suited for few observed tasks. I don't see either being demonstrated.

---

### Official Review · Reviewer_9uRZ · 2025-03-01

**Rating:** 6
**Confidence:** 4

**Review:**

### Summary
The paper introduces Sparse Gaussian Neural Processes, a novel approach that integrates sparse Gaussian processes (GPs) with the neural process (NP) model family, enabling fast meta-learning with interpretability and scalability. Specifically, the authors follow amortized variational inference, where a deep network (DeepSet/Transformer) first predicts variational parameters, which are then used as context points for a sparse GP. Additionally, the model incorporates an inductive bias by conditioning on a latent representation from a ConvDeepSet, utilizing a kvv parameterization trick to enforce structure in the variational posterior. The authors demonstrate the model’s effectiveness on both synthetic and real-world datasets.

### Strengths
- By amortizing variational inference, SGNP infers a sparse GP posterior in a single forward pass, significantly reducing computational cost compared to standard GP-based meta-learning.
- SGNP exhibits strong data efficiency, outperforming ConvGNP in low-data regimes while maintaining competitive performance with larger meta-training datasets.
- The ability to incorporate domain knowledge via kernel priors (as demonstrated in the Tétouan power consumption dataset) enhances practical relevance, making SGNP more interpretable than black-box NPs.
- The method is well-grounded in variational GP inference, ensuring mathematical soundness.

### Weaknesses & Questions
- While SGNP improves interpretability and scalability, it introduces higher model complexity, requiring multiple neural network forward passes and a GP inference step, making implementation and tuning more challenging than simpler GP or NP models. This also increases the number of hyperparameters to tune.
- The method appears to assume shared kernel parameters across tasks, which may limit adaptability to heterogeneous task distributions and diminish the benefits of DeepSet/Transformer architectures.
- The experimental evaluation focuses only on low-dimensional settings (1D/2D regression and classification). How well does SGNP scale to higher-dimensional tasks such as image or multi-output function modeling?

### Suggestions
- The authors primarily use negative log-likelihood (NLL) as the evaluation metric. Would incorporating L2 loss for regression tasks provide additional insights into the model’s effectiveness, particularly in capturing predictive mean accuracy?
- The experiments do not include deep learning NP baselines such as vanilla NP or Attentive NP (ANP). Could the authors demonstrate how these models fail in low-data scenarios, further highlighting the advantages of SGNP?

---

### Official Review · Reviewer_DP5b · 2025-03-01
**This paper presents a meta-learning method that combines sparse Gaussian processes and neural processes. The evaluation is done using two synthetic datasets, one for regression and the other one for classification, and one real-world dataset.**

**Rating:** 4
**Confidence:** 4

**Review:**

There is large room to improve in the writing.  I do not think this paper is acceptable with its current writing quality. Below are a few examples.
•	There are multiple components involved in the proposed method. The authors are suggested to have a diagram to illustrate their method.
•	q(u) is determined by m and S that are outputs of neural networks. I do not see how prior of any is involved in the calculation.
•	Their overall learning setup is unclear to me. If there is no data portioning into context and target, how predicting for unseen datasets was done?
•	In Eq. (17), what is k? index of tasks? Also, what is D_c? the context dataset? But it was mentioned by the authors that there is no partitioning of data into context and target subsets.

On overall, the work appears incremental, a straightforward assembly of different things, lacking technical innovation.

The empirical results are not convincing. It appears to me ConvSGNP is their full model while the SGNP is a tailored version. Why ConvSGNP performs much worse than SGNP in Table 1? I cannot find any discussion about this. In addition, ConvSGNP performed much worse than ConvGNP in the same table, which suggests the proposed method is not working as they thought. Also, in their real-world data experiment, the proposed method is not the best performing one. It is difficult for me to see the true value of their method. Moreover, their empirical study lacks comparison with the vast majority of existing meta-learning / zero-shot learning methods in the literature.  Finaly, the method should be evaluated on more real-world datasets.

---

### Official Review · Reviewer_jqrc · 2025-03-02

**Rating:** 6
**Confidence:** 3

**Review:**

# Summary

This work is an incremental yet meaningful improvement over the existing Gaussian neural process models used in probabilistic meta-learning in terms of efficiency and scalability. They try this method on a real-world task as well as a synthetic classification and regression task and shows that it outperform existing Gaussian neural process methods.

# Strengths

1. The paper bridges the gap between the flexibility of neural networks and the interpretability and efficiency of Gaussian processes (GPs)
2. Sparse GPs in a meta-learning setting is a new and useful approach.
3. Solid theoretical foundation
4. The experiments demonstrate that this method outperform the existing neural process methods, especially when the number of data points are small.
5. The contributions to probabilistic meta-learning are significant in terms of scalability, efficiency, and interpretability.

# Weakness

1. The authors claim improved efficiency through sparse GPs compared to GPs. A further computational complexity analysis comparing this work SGNP against Non-GP meta-learning models, such as transformer-based models, would be interesting to discuss.
2. Following the weakness 2, it is unclear about the instances where SGNP is preferable over Gaussian process or deep meta learning models.
3. The claim that the model is scalable needs to be proved by trying this method on further larger-scale benchmarks

---

### Meta-Review · Area_Chair_JZcy · 2025-03-17

**Recommendation:** Accept
**Confidence:** 3

**Metareview:**

The authors propose a new meta-learning approach based on sparse Gaussian neural process. The model leads to a more computationally and data efficient meta-learner. The reviews for this paper were borderline, however I find it interesting and relevant for the community. I recommend acceptance as long as the promised revisions to the paper and new baselines/experiments promised in the rebuttal are included in the camera-ready.

---

### Decision · Program_Chairs · 2025-03-18

**Decision:**

Accept

**Comment:**

Accept conditional on promised revisions in the camera ready.